META-RESEARCH ARTICLE

# Linking citation and retraction data reveals the demographics of scientific retractions among highly cited authors

John P. A. Ioannidis[1,2,3,4,5]*, Angelo Maria Pezzullo[5,6], Antonio Cristiano[5,6], Stefania Boccia[6], Jeroen Baas[7]

1 Department of Medicine, Stanford University, Stanford, California, United States of America, 2 Department of Epidemiology and Population Health, Stanford University, Stanford, California, United States of America, 3 Department of Biomedical Data Science, Stanford University, Stanford, California, United States of America, 4 Stanford Center for Innovation in Global Health, Stanford University, Stanford, California, United States of America, 5 Meta-Research Innovation Center at Stanford (METRICS), Stanford University, Stanford, California, United States of America, 6 Section of Hygiene, Department of Life Sciences and Public Health, Università Cattolica del Sacro Cuore, Rome, Italy, 7 Research Intelligence, Elsevier B.V., Amsterdam, the Netherlands

* jioannid@stanford.edu

**Data Availability Statement:** The full datasets are available at https://doi.org/10.17632/btchxktzyw.7.

## Abstract

Retractions are becoming increasingly common but still account for a small minority of published papers. It would be useful to generate databases where the presence of retractions can be linked to impact metrics of each scientist. We have thus incorporated retraction data in an updated Scopus-based database of highly cited scientists (top 2% in each scientific subfield according to a composite citation indicator). Using data from the Retraction Watch database (RWDB), retraction records were linked to Scopus citation data. Of 55,237 items in RWDB as of August 15, 2024, we excluded non-retractions, retractions clearly not due to any author error, retractions where the paper had been republished, and items not linkable to Scopus records. Eventually, 39,468 eligible retractions were linked to Scopus. Among 217,097 top-cited scientists in career-long impact and 223,152 in single recent year (2023) impact, 7,083 (3.3%) and 8,747 (4.0%), respectively, had at least 1 retraction. Scientists with retracted publications had younger publication age, higher self-citation rates, and larger publication volume than those without any retracted publications. Retractions were more common in the life sciences and rare or nonexistent in several other disciplines. In several developing countries, very high proportions of top-cited scientists had retractions (highest in Senegal (66.7%), Ecuador (28.6%), and Pakistan (27.8%) in career-long citation impact lists). Variability in retraction rates across fields and countries suggests differences in research practices, scrutiny, and ease of retraction. Addition of retraction data enhances the granularity of top-cited scientists' profiles, aiding in responsible research evaluation. However, caution is needed when interpreting retractions, as they do not always signify misconduct; further analysis on a case-by-case basis is essential. The database should hopefully provide a resource for meta-research and deeper insights into scientific practices.

**Funding:** The work of AC has been supported by the European Network Staff Exchange for Integrating Precision Health in the Healthcare Systems project (Marie Skłodowska-Curie Research and Innovation Staff Exchange no. 823995). The funders had no role in study design, data collection and analysis, decision to publish, or preparation of the manuscript.

**Competing interests:** JB is an Elsevier employee. Elsevier runs Scopus, which is the source of these data, and also runs the repository where the database of highly-cited scientists is now stored.

## Introduction

Retractions of publications are a central challenge for science and their features require careful study [1–3]. In empirical surveys, various types of misconduct are typically responsible for most retractions [4]. The landscape of retractions is becoming more complex with the advent of papermills, massive production of papers that are typically fake/fabricated and where people may buy authorship in their masthead [5]. However, the reasons for retractions are not fully standardized, and many retractions are unclear about why a paper had to be withdrawn. Moreover, some retractions are clearly not due to ethical violations or author errors (e.g., they are due to publisher errors). Finally, in many cases, one may view a retraction as a sign of a responsible author who should be congratulated, rather than chastised, for taking proactive steps to correct the literature. Prompt correction of honest errors, major or minor, is a sign of responsible research practices.

The number of retracted papers per year is increasing, with more than 10,000 papers retracted in 2023 [6]. The countries with the highest retraction rates (per 10,000 papers) are Saudi Arabia (30.6), Pakistan (28.1), Russia (24.9), China (23.5), Egypt (18.8), Malaysia (17.2), Iran (16.7), and India (15.2) [6]. However, retractions abound also in highly developed countries [7]. There has also been a gradual change in the reasons for retractions over time [8]: the classic, traditional types of research misconduct (falsification, fabrication, plagiarism, and duplication) that involved usually one or a few papers at a time have been replaced in the top reasons by large-scale, orchestrated fraudulent practices (papermills, fake peer-review, artificial intelligence generated content). Clinical and life sciences account for about half of the retractions that are apparently due to misconduct [9], but electrical engineering/electronics/computer science (EEECS) have an even higher proportion of retractions per 10,000 published papers [9]. Clinical and life sciences disciplines have the highest rates of retractions due to traditional reasons of misconduct, while EEECS disciplines have a preponderance of large-scale orchestrated fraudulent practices.

Here, we aimed to analyze the presence of any retracted papers for all the top-cited scientists across all 174 subfields of science. Typical impact metrics for scientists revolve around publications and their citations. However, citation metrics need to be used with caution [10] to avoid obtaining over-simplified and even grossly misleading views of scientific excellence and impact. We therefore updated and extended databases of standardized citation metrics across all scientists and scientific disciplines [11–14] to include information on retractions for each scientist. Systematic indicators of research quality and integrity are important to examine side-by-side with traditional citation impact data [15,16]. A widely visible list of highly cited scientists issued annually by Clarivate based on Web of Science no longer includes any scientists with retracted publications [17]. In our databases, which cover a much larger number of scientists with more detailed data on each, we have added information on the number of retracted publications, if any, for all listed scientists. Given the variability of the reasons behind retraction, this information can then be interpreted by any assessors on a case-by-case basis with in-depth assessment of reasons, and circumstances of each retraction.

Using our expanded databases, we aimed to answer the following questions: How commonly have top-cited scientists retracted papers? Are there any features that differentiate top-cited scientists with versus without retracted papers? Are specific scientific fields and subfields more likely to have top-cited scientists with retracted papers? Do some countries have higher rates of retractions among their top-cited scientists? Finally, how much do citations to and from retracted papers contribute to the overall citation profile of top-cited scientists? As we present these analyses, we also hope that this new resource will be useful for further meta-research studies that may be conducted by investigators on diverse samples of scientists and scientific fields.

## Methods and results

To add the new information on retractions, we depended on the most reliable database of retractions available to date, the Retraction Watch database (RWDB, RRID:SCR_000654) which is also publicly freely available through CrossRef (RRID:SCR_003217). Among the 55,237 RWDB entries obtained from CrossRef (https://api.labs.crossref.org/data/retractionwatch) on August 15, 2024, we focused on the 50,457 entries where the nature of the notice is classified as "Retraction", excluding other types (corrections, expressions of concern) that may also be covered in RWDB. From this set, we excluded entries where the paper had been retracted but then replaced by a new version (which can suggest that the errors were manageable to address and there is a new version representing the work in the published literature), and those entries where the retraction was clearly solely not due to any error or wrongdoing by the authors (e.g., publisher error). Therefore, we excluded entries where the reason for retraction was listed as "Retract and Replace," "Error by Journal/Publisher," "Duplicate Publication through Error by Journal/Publisher," or "Withdrawn (out of date)"; however, for the latter 3 categories, these exclusions were only applied if there were no additional reasons listed that could be attributed potentially to the authors exclusively or in part, as detailed in S1 Table. This first filtering was automated and resulted in a set of 47,964 entries.

We tagged articles as retracted by linking retraction records to their corresponding entries in Scopus (RRID:SCR_022559). Initially, this linking is achieved by matching the OriginalPaperDOI with a DOI in Scopus. For retracted articles that do not have a direct DOI match, we employ an alternative strategy using the title and publication year, allowing for a 1-year discrepancy due to variations in the recorded publication year. To enhance the accuracy of our linking process, we perform data sanitization on both databases. DOIs are standardized by removing redundant prefixes and extraneous characters. Titles are normalized by stripping all non-alphanumeric characters and converting them to lowercase. Additionally, to avoid erroneous matches, especially with shorter titles, we impose a minimum length requirement of 32 characters for title matching. The code that demonstrates the linking strategy is published along the data set at https://elsevier.digitalcommonsdata.com/datasets/btchxktzyw/7.

Linking the retraction using the digital object identifier (DOI) of the original paper resulted in 38,364 matches. For entries where a DOI match was not possible, and where we attempted to link records using a combination of the title and year derived from the date of the original article, allowing for a +/− 1-year variation, resulted in 1,104 additional matches. This linkage process eventually resulted in a total of 39,468 matched records (Fig 1).

Calculation of the composite citation indicator and ranking of the scientists accordingly within their primary subfield (using the Science-Metrix classification of 20 fields and 174 subfields) were performed in the current iteration with the exact same methods as in previous iterations (described in detail in references [11–13]). Career-long impact counts citations received cumulatively across all years to papers published at any time, while single most recent year impact counts only citations received in 2023 to papers published at any time.

The new updated release of the databases includes 217,097 scientists who are among the top 2% of their primary scientific subfield in the career-long citation impact and 223,152 scientists who are among the top 2% in their single most recent year (2023) citation impact. These numbers also include some scientists (2,789 and 6,325 scientists in the 2 data sets, respectively) who may not be in the top 2% of their primary scientific subfield but are among the 100,000 top-cited across all scientific subfields combined. Among the top-cited scientists, 7,083 (3.3%) and 8,747 (4.0%), respectively, in the 2 datasets have at least 1 retracted publication, and 1,710 (0.8%) and 2,150 (1.0%), respectively, have 2 or more retracted publications. As shown in Fig 2, the distribution of the number of linked eligible retractions per author follows a power law.

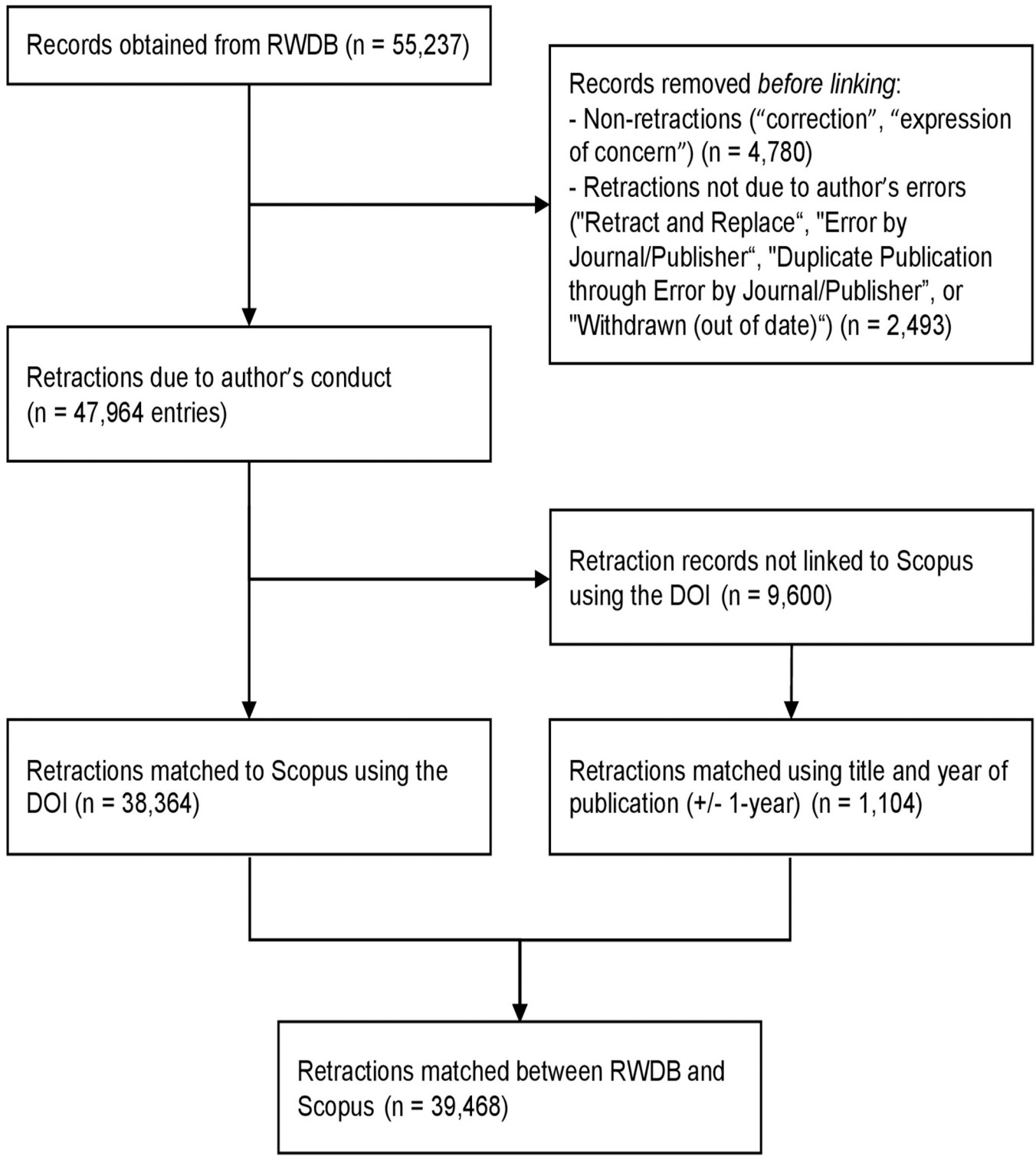

**Fig 1. Flow diagram for linkage of retractions.**

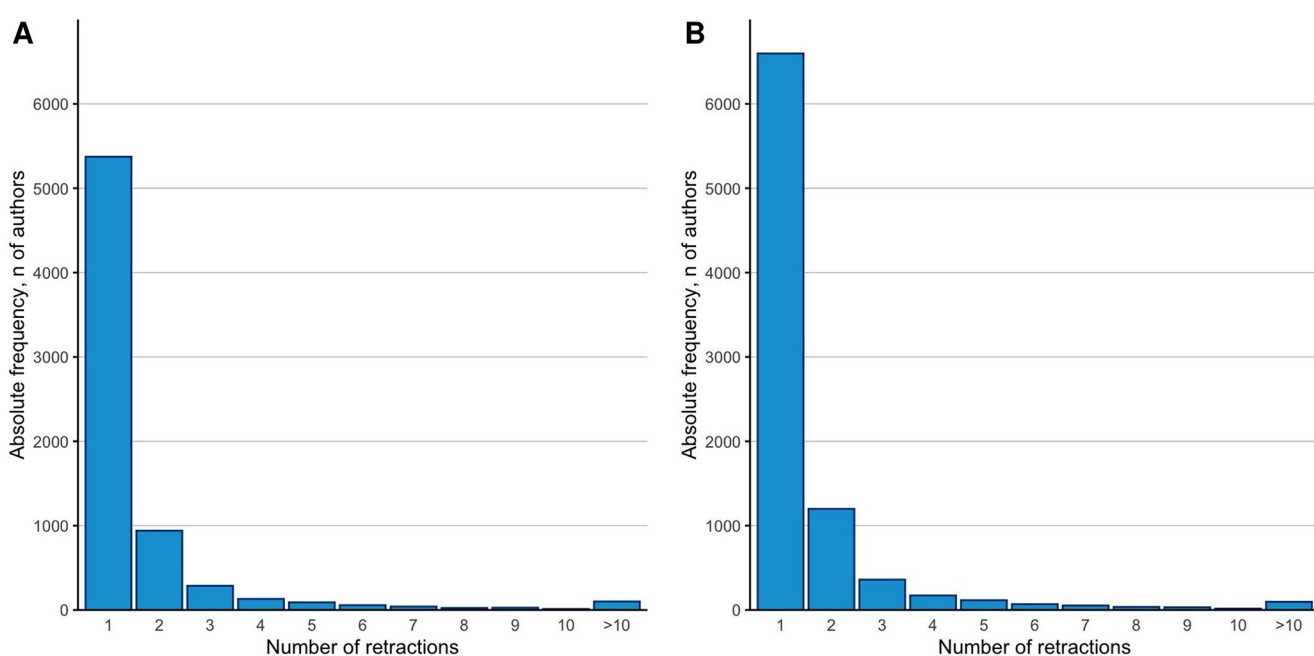

**Fig 2. Distribution of the number of retractions in top-cited scientists with at least 1 retraction.** (A) Database of top-cited authors based on career-long impact. (B) Database of top-cited authors based on single recent year (2023) impact. The data underlying this figure can be found in S1 Data.

Table 1 shows the characteristics of those top-cited scientists who have any retracted publications versus those who have not had any retractions. As shown, top-cited scientists with retracted publications tend to have younger publication ages, higher proportion of self-citations, higher ratio of h/hm index (indicating higher co-authorship levels), slightly better ranking, and higher total number of publications ($p < 0.001$ by Mann–Whitney $U$ test (in R version 4.4.0 (RRID: SCR_001905))) for all indicators in the career-long impact data set and the single recent year data set, except for the publication age and the absolute ranking in the subfield in the single recent year data set. However, except for the number of papers published, the differences are small or modest in absolute magnitude. The proportion of scientists with retractions is highest though at the extreme top of ranking. Among the top 1,000 scientists

**Table 1. Top-cited scientists with and without retracted publications characteristics and Mann–Whitney $U$ test.**

|  | Career-long impact | | | Single recent year (2023) impact | | |
|---|---|---|---|---|---|---|
|  | **Retracted** | **Others** |  | **Retracted** | **Others** |  |
|  | **N = 7,083** | **N = 210,014** | **p-value** | **N = 8,747** | **N = 214,405** | **p-value** |
| Publication start, median (IQR) | 1989 (1981–1997) | 1987 (1977–1996) | <0.00001 | 1997 (1987–2005) | 1997 (1987–2006) | 0.2 |
| Self-citations (%), median (IQR) | 12.9 (9.6–17.6) | 11.7 (7.5–16.9) | <0.00001 | 9.1 (5.6–14) | 8.8 (4.8–14.2) | <0.00001 |
| h-index/hm-index ratio*, median (IQR) | 2.4 (2–2.8) | 2.1 (1.7–2.6) | <0.00001 | 2.1 (1.8–2.6) | 2 (1.7–2.5) | <0.00001 |
| Ranking in subfield, median (IQR) | 973 (342–2,128.5) | 1,011 (381–2,150) | 0.0007 | 1,029 (367–2,274) | 1,025 (388–2,170) | 0.69 |
| Percentile ranking in subfield, median (IQR) | 0.008 (0.003–0.014) | 0.011 (0.005–0.016) | <0.00001 | 0.009 (0.003–0.015) | 0.011 (0.006–0.016) | <0.00001 |
| Number of total published items, median (IQR) | 270 (170–426) | 160 (100–253) | <0.00001 | 228 (135–377) | 139 (79–234) | <0.00001 |

* Data on h-index/hm-index are including self-citations. The Schreiber hm index is constructed in the same way as the Hirsch h-index but considers also co-authorship. The more extensive the co-authorship, the more the hm index will deviate from (and become smaller than) the h-index.

**Table 2. Top-cited scientists with and without/ retracted publications according to their main field.**

| Main field | Career-long impact | | Single recent year impact | |
|---|---|---|---|---|
| | Retracted | Others | Retracted | Others |
| | *N* = 7,083 | *N* = 210,014 | *N* = 8,747 | *N* = 214,405 |
| Agriculture, Fisheries & Forestry | 99 (1.4%) | 7,166 (98.6%) | 172 (2.3%) | 7,203 (97.7%) |
| Biology | 222 (2.6%) | 8,434 (97.4%) | 300 (3.5%) | 8,363 (96.5%) |
| Biomedical Research | 846 (5.0%) | 16,052 (95.0%) | 847 (5.1%) | 15,843 (94.9%) |
| Built Environment & Design | 34 (2.7%) | 1,209 (97.3%) | 40 (3.1%) | 1,263 (96.9%) |
| Chemistry | 462 (3.1%) | 14,449 (96.9%) | 624 (4.1%) | 14,565 (95.9%) |
| Clinical Medicine | 3,249 (4.8%) | 64,590 (95.2%) | 3,769 (5.5%) | 64,574 (94.5%) |
| Communication & Textual Studies | 2 (0.2%) | 1,072 (99.8%) | 4 (0.3%) | 1,193 (99.7%) |
| Earth & Environmental Sciences | 157 (2.1%) | 7,231 (97.9%) | 216 (2.8%) | 7,526 (97.2%) |
| Economics & Business | 59 (1.4%) | 4,078 (98.6%) | 111 (1.8%) | 6,171 (98.2%) |
| Enabling & Strategic Technologies | 654 (3.6%) | 17,663 (96.4%) | 906 (4.4%) | 19,790 (95.6%) |
| Engineering | 432 (2.5%) | 16,686 (97.5%) | 565 (3.3%) | 16,631 (96.7%) |
| Historical Studies | 0 (0.0%) | 1,081 (100.0%) | 1 (0.1%) | 1,073 (99.9%) |
| Information & Communication Technologies | 275 (1.8%) | 14,812 (98.2%) | 475 (3.1%) | 14,700 (96.9%) |
| Mathematics & Statistics | 48 (1.8%) | 2,645 (98.2%) | 80 (2.9%) | 2,639 (97.1%) |
| Philosophy & Theology | 0 (0.0%) | 523 (100.0%) | 2 (0.4%) | 524 (99.6%) |
| Physics & Astronomy | 361 (1.8%) | 19,619 (98.2%) | 431 (2.3%) | 18,576 (97.7%) |
| Psychology & Cognitive Sciences | 98 (2.5%) | 3,773 (97.5%) | 108 (2.6%) | 4,036 (97.4%) |
| Public Health & Health Services | 65 (1.7%) | 3,776 (98.3%) | 66 (1.7%) | 3,803 (98.3%) |
| Social Sciences | 20 (0.4%) | 5,043 (99.6%) | 30 (0.5%) | 5,818 (99.5%) |
| Visual & Performing Arts | 0 (0.0%) | 112 (100.0%) | 0 (0.0%) | 114 (100.0%) |

with the highest composite indicator values, the proportion of those with at least 1 retraction are 13.8% and 11.1%, in the career-long and single recent year impact, respectively.

Table 2 shows the proportion of top-cited scientists with retracted publications across the 20 major fields that science is divided according to the Science-Metrix classification; information on the more detailed 174 subfields appears in S2 Table. The proportion of retractions varies widely across major fields, ranging from 0% to 5.5%. Clinical Medicine and Biomedical Research have the highest rates (4.8% to 5.5%). Enabling & Strategic Technologies, Chemistry and Biology have rates close to the average of all sciences combined. All other fields have from low to very low (or even zero) rates of scientists with retractions. When the 174 Science-Metrix subfields of science were considered, the highest proportions of top-cited scientists with at least 1 retracted paper were seen in the subfields of Complementary & Alternative Medicine, Oncology & Carcinogenesis, and Pharmacology & Pharmacy (with 10.5%, 9.9%, and 9.4%, respectively of top-cited scientists based on single recent year impact). See details on all 174 subfields in S2 Table.

Retraction rates among top-cited scientists also vary in the 20 countries that host most of the top-cited authors (Table 3), with higher rates observed in India (9.2% career-long to 8.6% single recent year impact), China (8.2% to 6.7%), and Taiwan (5.2% to 5.7%), and lower rates observed in Israel (1.7% to 2.0%), Belgium (2.1% to 2.1%), and Finland (2.2% to 2.2%). Some countries with few top-cited authors (not among the 20 shown in Table 2) have impressive rates of scientists with retractions: Countries that exceed 10% either in career-long or in single recent year top-cited scientists are listed in S3 Table. The highest proportions of top-cited scientists with retractions were seen in Senegal (66.7%), Ecuador (28.6%), and Pakistan (27.8%) based on the career-long impact list and in Kyrgyzstan (50%), Senegal (41.7%), Ecuador

**Table 3. Top-cited scientists with and without retracted publications according to country.**

| Country | Career-long impact | | Single recent year impact | |
|---|---|---|---|---|
| | Retracted | Others | Retracted | Others |
| United States | 2,332 (2.8%) | 81,870 (97.2%) | 2,186 (3.1%) | 69,206 (96.9%) |
| United Kingdom | 430 (2.2%) | 19,218 (97.8%) | 428 (2.4%) | 17,127 (97.6%) |
| Germany | 336 (2.9%) | 11,236 (97.1%) | 309 (3.0%) | 10,111 (97.0%) |
| China | 877 (8.2%) | 9,810 (91.8%) | 1,813 (6.7%) | 25,352 (93.3%) |
| Canada | 241 (2.6%) | 9,024 (97.4%) | 223 (2.7%) | 7,962 (97.3%) |
| Japan | 362 (4.4%) | 7,899 (95.6%) | 254 (4.5%) | 5,354 (95.5%) |
| Australia | 178 (2.4%) | 7,270 (97.6%) | 201 (2.5%) | 7,833 (97.5%) |
| France | 151 (2.2%) | 6,770 (97.8%) | 152 (2.6%) | 5,630 (97.4%) |
| Italy | 254 (4.1%) | 6,017 (95.9%) | 300 (3.9%) | 7,318 (96.1%) |
| Netherlands | 123 (2.7%) | 4,392 (97.3%) | 116 (2.6%) | 4,419 (97.4%) |
| Spain | 103 (2.9%) | 3,405 (97.1%) | 127 (3.2%) | 3,880 (96.8%) |
| Switzerland | 84 (2.4%) | 3,347 (97.6%) | 82 (2.4%) | 3,323 (97.6%) |
| Sweden | 84 (2.5%) | 3,269 (97.5%) | 78 (3.0%) | 2,566 (97.0%) |
| India | 270 (9.2%) | 2,669 (90.8%) | 462 (8.6%) | 4,889 (91.4%) |
| South Korea | 120 (5.1%) | 2,246 (94.9%) | 186 (5.3%) | 3,313 (94.7%) |
| Denmark | 46 (2.2%) | 2,068 (97.8%) | 53 (2.6%) | 1,960 (97.4%) |
| Israel | 36 (1.7%) | 2,057 (98.3%) | 32 (2.0%) | 1,590 (98.0%) |
| Belgium | 41 (2.1%) | 1,956 (97.9%) | 42 (2.1%) | 1,965 (97.9%) |
| Taiwan | 91 (5.2%) | 1,668 (94.8%) | 80 (5.7%) | 1,327 (94.3%) |
| Finland | 32 (2.2%) | 1,413 (97.8%) | 26 (2.2%) | 1,153 (97.8%) |

(28%), and Belarus (26.7%) in the single recent year impact list. Nevertheless, the total number of top-cited authors for Senegal, Ecuador, Kyrgyzstan, and Belarus is very small, so percentages should be seen with caution.

The new iteration of the 2 top-cited scientists' data sets also includes information on the number of citations received (overall and in the single recent year, respectively) by the retracted papers of each scientist. If we consider scientists with at least 1 retraction, the range is 0 to 7,491, with median (IQR) of 25 (6 to 80) in the career-long data set. The range is 0 to 832 with median (IQR) of 1 (0 to 4) in the single recent year data set. A total of 114 scientists in the career-long data set have received more than 1,000 citations to their retracted papers and for 230 (0.1%) and 260 (0.1%) scientists in the 2 data sets the citations to the retracted papers account for more than 5% of their citations.

Furthermore, information is provided for each scientist on the number of citations that they received from any of the retracted papers. In the career-long data set, the range is 0 to 1,974, with median (IQR) of 0 (2 to 5) and in the single recent year data set, the range is 0 to 180 with median (IQR) of 0 (0 to 0). A total of 5 scientists in the career-long data set have received more than 1,000 citations from papers that have been retracted and for 14 and 7 scientists in the 2 data sets, the citations they have received from retracted papers account for more than 5% of their citations (overall and in the single recent year, respectively).

## Discussion

We hope that the addition of the retraction data will improve the granularity of the information provided on each scientist in the new, expanded database of top-cited scientists. A more informative profile may be obtained by examining not only the citation indicators but retracted papers, proportion of self-citations, evidence of extremely prolific behavior [18] (see

detailed data that can be linked to the top-cited scientists' database, published in https://elsevier.digitalcommonsdata.com/datasets/kmyvjk3xmd/2), as well as responsible indicators such as data and code sharing and protocol registration information that is becoming increasingly available [15,16].

The data suggest that approximately 4% of the top-cited scientists have at least 1 retraction. This is a conservative estimate, and the true rate may be higher since some retractions are in titles that are not covered by Scopus or could not be linked in our data set linkage. Proportions of scientists with retractions are substantially higher in the extremes of the most-cited scientists. Top-cited scientists with retracted publications exhibit higher levels of collaborative co-authorship and have a higher total number of papers published. High productivity and more extensive co-authorship may be associated with less control over what gets published or may show proficiency in gaming the system (e.g., have honorary authorship as department chair). Nevertheless, the higher publication output of scientists with retractions might simply reflect that the more you publish, the greater the chance of encountering eventually a retraction.

More than half of the top-cited scientists with retractions were in medicine and life sciences. However, high rates were seen also in several other fields. A previous mapping of retractions due to misconduct [9] had found the highest rates of retracted papers in EEECS at 18 per 10,000, double the rate for life sciences. The EEECS scientific area corresponds in our mapping to scientific domains where we also found high concentrations of top-cited scientists with retracted papers, although the rates were lower than the rate in Clinical Medicine and in Biomedical Research. It is possible that medical and life science retractions are more likely to involve top-cited authors, while EEECS retracted papers have mostly authors who do not manage to reach top-cited status. EEECS retractions have a large share of artificial intelligence generated content and fake review [9]. Therefore, it is likely that such fraudsters aim for more modest citation records, or they are revealed before they reach highly cited status, although exceptions do exist [9]. Many scientific fields have minimal or no track records of retractions and some subfields such as alternative medicine, cancer research, and pharmacology exhibit rates of retractions double the rates exhibited by the life sciences overall. These differences might reflect the increased scrutiny and better detection of misconduct and major errors in fields that have consequences for health; differences in the intensity and types of post-publication review practices [19]; and the fact that quantifiable data and images in the life sciences are easier to assess for errors and fraud than many constructs in social sciences.

Many developing countries have extremely high rates of top-cited authors with retracted papers. This may reflect problematic research environments and incentives in these countries, several of which are also rapidly growing their overall productivity [3,20–23]. The countries where we detected the highest rates of top-cited authors with retractions largely overlap with the countries that have also the highest number of retracted papers per 10,000 publications according to a previous mapping of retractions due to misconduct [9]. In fact, some of these countries such as India, China, Pakistan, and Iran also have a large share of implausibly hyperprolific authors [18]. It would be interesting to see if removing some of the productivity incentives may reduce the magnitude of the problem in these countries.

As previously documented, several retracted papers have been cited considerably and, unfortunately, some continue to be cited even after their retraction [24,25]—these citations are typically such that they suggest that the citing authors are unaware of the retraction rather than citing the paper to comment on its retraction. This is a problem that should and can be hopefully fixed.

Among top-cited authors, a small number have received a very large number of citations to their retracted papers. However, these citations have a relatively small proportional contribution to the overall very high total citation counts of these scientists. The same applies to the proportion of citations that are received by retracted papers. Some highly cited authors may

have received a substantial number of citations from retracted papers, but this is a very small proportion against their total citations. Nevertheless, within paper mills, fake papers may be using repeatedly the same citations from known, influential authors and papers that are already cited heavily in the literature. It is possible that most paper mill products remain undetected and have not yet been retracted from the literature.

We expect that the new, expanded database may enhance the progression of further research on citation and retraction indicators, with expanded linkage to yet more research indicators. We caution that even though we excluded retractions that attributed no fault to the authors, we cannot be confident that all the included retractions included some error, let alone misconduct, by the authors. Some retraction notes are vague and the separation of author-related versus author-unrelated reasons may not be perfect. Even for types of reasons that seem to be author-related, exceptions may exist, e.g., in partial fake review, it could be that the editors unexpectedly invite a fake referee or encounter review mills [26]. Moreover, sometimes not all authors may have been responsible for what led to the retraction. Therefore, any further analyses that focus on individual author profiles rather than aggregate, group-level analyses should pay due caution in dissecting the features and circumstances surrounding each retraction. Unfortunately, these are often not presented in sufficient detail to allow safe judgments [4,27].

Moreover, inaccuracies are possible in the merged data set. As discussed previously, Scopus has high precision and recall [28], but some errors do exist in author ID files. In the past, some Asian author IDs had very high numbers of papers because more than one author were merged in the same ID file. However, this is no longer the case and Asian name disambiguation in Scopus is currently as good or even better than European/American names [28]. Errors may also happen in the attribution of affiliation for each scientist. Finally, considering the vast size of these data sets with potential duplicity and similarity of names, ensuring that no scientist is incorrectly associated with a retracted paper is virtually impossible. Users of these data sets and/or Scopus can improve author profile accuracy by offering corrections directly to Scopus through the use of the Scopus to ORCID feedback wizard (https://orcid.scopusfeedback.com/). Most importantly, we make no judgment calls in our databases on the ethical nature of the retractions, e.g., whether they represent misconduct or honest error by the authors; similarly we do not comment on whether the retractions may be fair or not. Some retractions may still be contested between authors and editors and/or may even have ongoing legal proceedings. We urge users of these data to very carefully examine the evidence and data surrounding each retraction and its nature.

## Supporting information

**S1 Data. Data underlying Fig 2.**
(DOCX)

**S1 Table. List of author-attributable reasons used to filter journal error and withdrawn (out of date) exceptions.**
(DOCX)

**S2 Table. Top-cited scientists with and without retracted publications according to their primary subfield.**
(DOCX)

**S3 Table. Top-cited scientists with and without retracted publications in countries with high (>10%) retraction prevalence.**
(DOCX)

## Acknowledgments

This work uses Scopus data provided by Elsevier. We are thankful to Alison Abritis and Ivan Oransky for constructive comments.

## Author Contributions

**Conceptualization:** John P. A. Ioannidis, Angelo Maria Pezzullo, Stefania Boccia, Jeroen Baas.

**Data curation:** John P. A. Ioannidis, Angelo Maria Pezzullo, Antonio Cristiano, Jeroen Baas.

**Formal analysis:** Angelo Maria Pezzullo, Antonio Cristiano, Jeroen Baas.

**Investigation:** John P. A. Ioannidis, Jeroen Baas.

**Methodology:** John P. A. Ioannidis, Jeroen Baas.

**Resources:** Stefania Boccia.

**Supervision:** John P. A. Ioannidis, Stefania Boccia.

**Validation:** John P. A. Ioannidis, Angelo Maria Pezzullo, Antonio Cristiano, Jeroen Baas.

**Visualization:** Angelo Maria Pezzullo.

**Writing – original draft:** John P. A. Ioannidis.

**Writing – review & editing:** John P. A. Ioannidis, Angelo Maria Pezzullo, Antonio Cristiano, Stefania Boccia, Jeroen Baas.

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
