## [Editor Report · Decision Letter 0]

30 Sep 2024

Dear John, 

Thank you for submitting your manuscript entitled "Updated science-wide author databases of standardized citation indicators including retraction data" for consideration as a Meta-Research Article by PLOS Biology.

Your manuscript has now been evaluated by the PLOS Biology editorial staff, as well as by an academic editor with relevant expertise, and I'm writing to let you know that we would like to send your submission out for external peer review.

IMPORTANT: In order to help maximise our chances of recruiting appropriate reviewers (and probably maximising the chances of a positive outcome?), we strongly suggest that you slightly re-frame the article before uploading the additional metadata (see next paragraph). SPECIFICALLY, while we recognise the popularity of your database, we think that it would be better to lead with the retraction analysis, leaving the database update to be a secondary aspect. I think this could be relatively easily done, with a tweak to the Title, and then re-ordering the relevant elements of the Abstract and Introduction. I don't think any changes would be needed in the rest of the manuscript. In answer to your question about Meta-Research Article versus Update Article, we would definitely keep it as a Meta-Research Article. If the afore-mentioned re-framing is likely to take more than a week, let me know, and we can "reject" and then allow a new submission when you're ready (simply a formality).

Once your full submission is complete, your paper will undergo a series of checks in preparation for peer review. After your manuscript has passed the checks it will be sent out for review. To provide the metadata for your submission, please Login to Editorial Manager (https://www.editorialmanager.com/pbiology) within two working days, i.e. by Oct 02 2024 11:59PM.

Kind regards,

Roli

Roland Roberts, PhD

Senior Editor

PLOS Biology

rroberts@plos.org

---

## [Decision Letter · Decision Letter 1]

15 Nov 2024

Dear John,

Thank you for your patience while your manuscript "Retractions among highly-cited authors in science-wide author databases" went through peer-review at PLOS Biology. Your manuscript has now been evaluated by the PLOS Biology editors, an Academic Editor with relevant expertise, and by four independent reviewers.

You'll see that reviewer #1 says that the study is new and important, and simply has a few questions about methodology and a suggestions for some helpful diagrams. Reviewer #2 is also positive, but wants you to discuss more of the prior literature on retractions, to better justify a claim, and to point out that some retractions may not be down to the authors themselves. Reviewer #3 wants you to formulate clearer research questions, questions the point of discussing countries with very low publication rates, questions the rationale behind looking at citations in such a recent year as 2023, and wants more detail about name disambiguation. Reviewer #4 says that the paper is important, but thinks that it needs re-framing and clearer motivation (is it about the database, or is it about the retractions?), and wants more clarity on where the responsibility for retraction lies.

IMPORTANT: My diagnosis here is that the concerns raised by reviewer #3 and #4 are a natural consequence of my previous request that you do a "quick and dirty" cosmetic re-framing of the paper before review, as we were much more interested in the retraction analysis than in the database per se, and wanted the reviewers to focus on that aspect. The reviewers seem to detect the resulting disconnect, so I see revision as an opportunity for you to complete the process of re-framing around the retraction aspect (e.g. by including clear research questions, as the reviewers suggest). Obviously the other concerns raised by the reviewers should also be addressed (and for clarity, we're still interested in the updated database, but the retraction analysis should take centre stage).

In light of the reviews, which you will find at the end of this email, we are pleased to offer you the opportunity to address the comments from the reviewers in a revision that we anticipate should not take you very long. We will then assess your revised manuscript and your response to the reviewers' comments with our Academic Editor aiming to avoid further rounds of peer-review, although might need to consult with the reviewers, depending on the nature of the revisions.

**IMPORTANT - SUBMITTING YOUR REVISION**

*Resubmission Checklist*

*Published Peer Review*

*PLOS Data Policy*

*Blot and Gel Data Policy*

Sincerely,

Roli

Roland Roberts, PhD

Senior Editor

PLOS Biology

rroberts@plos.org

REVIEWERS' COMMENTS:

Reviewer #1:

[identifies himself as David B Resnik]

This article develops databases that links retraction and citation data. The database can be useful for future research on retractions. The study is well designed and executed. The information provided is new and important. I have just a few questions/suggestions.

1. How was the linkage achieved? Was this automated? Done manually by members of the research team reviewing articles? 

2. Is this database publicly available? Where? Searchable?

3. You might want to include some diagrams as visual aids to the reader, such as the steps in your search and linkage of records.

Reviewer #2:

In their study, Ioannidis et al. conducted a bibliometric analysis of retractions among highly-cited authors, highlighting a surprising ratio of highly-cited scientists having at least one retraction. This study is generally interesting and holds some value. However, the work does suffer from some issues.

Firstly, the article primarily focuses on the retractions of highly-cited authors, but the background is too brief to provide a comprehensive understanding of the retraction landscape. The authors should consider reviewing the latest relevant articles for a broader background, such as the following literatures, https://www.nature.com/articles/d41586-023-03974-8, 10.1007/s11192-024-04992-7, 10.2478/jdis-2024-0012, 10.1016/j.xinn.2024.100593, 10.1016/j.heliyon.2024.e38620, as well as more literatures. This could enhance the significance of their work.

Secondly, the authors claimed that retractions are more prevalent in the life sciences compared to other fields. The authors may not reach this conclusion based solely on the proportion of top-cited scientists with retracted publications. This requires literature support or a comparison of retraction rates. 

Thirdly, the criteria used to filter journal errors may be inadequate. For instance, instances of partial fake peer review may not necessarily be linked to the authors. It could be due to factors like editors unexpectedly inviting a fake referee or encountering review mills (see 10.1007/s11192-024-05125-w). Additionally, some reasons for retractions, such as Falsification/Fabrication of Data, Contamination of Cell Lines/Tissues, Contamination of Materials, Duplication of Text, among others, were not explicitly categorized as author errors.

Lastly, there are issues with the citation style in the references. For example, "Oransky I. Volunteer watchdogs pushed a small country up the rankings. Science (1979). 2018;362(6413):395" requires correction. Please review the references for accuracy.

Reviewer #3:

This manuscript describes an update of a dataset of highly cited authors. The update also includes the addition of the number of retracted papers and their number of citations by the highly cited authors. I'm missing the research question of this manuscript. It might help to formulate clear research questions, e.g., how high is the percentage of highly cited authors with retracted papers, or do the retracted papers make some authors highly cited? I don't see the connection to biology. Maybe, the authors can include a research question that is related to biology. More specific comments follow below.

The authors state in the abstract without providing any reason: "It would be useful to generate databases where the presence of retractions can be linked to impact metrics of each scientist." They continue: "We have thus incorporated retraction data in an updated a Scopus-based database of highly-cited scientists (top-2% in each scientific subfield according to a composite citation indicator)." This is the other way around compared to the preceding sentence.

Also, in the abstract, the authors state: "In several developing countries, very high proportions of top-cited scientists had retractions (highest in Senegal (66.7%), Ecuador (28.6%) and Pakistan (27.8%) in career-long citation impact lists). Variability in retraction rates across fields and countries suggests differences in research practices, scrutiny, and ease of retraction." Especially, in the case of Senegal and Ecuador, this is statistics on very small numbers.

I do not see the benefit of analyzing the single most recent year (i.e., 2023) as this year's publications have had far too few time to be cited and generate impact.

The h/hm index has been mentioned (e.g., on page 6) but was not explained.

On page 9, the authors state: "Many developing countries have extremely high rates of top-cited authors with retracted papers. This may reflect problematic research environments and incentives in these countries, several of which are also rapidly growing their overall productivity (3,16-19). In fact, some of these countries such as India, China, Pakistan and Iran also have a large share of implausibly hyperprolific authors (14). It would be interesting to see if removing some of the productivity incentives may reduce the magnitude of the problem in these countries." I wonder if part of the "implausibly hyperprolific authors" and "extremely high rates of top-cited authors with retracted papers" might be due to a problem of the author name disambiguation. Also, no details regarding the author name disambiguation method were provided, like for the indicators used for ranking scholars.

Reviewer #4:

[identifies himself as Sean C. Rife]

The submitted manuscript investigates the increasing prevalence of retractions in scientific literature, which, despite their growth, still represent a small fraction of published works. By linking retraction data from the Retraction Watch database with citation metrics from Scopus, the authors found that a notable percentage of highly cited scientists had at least one retraction, with notable variations across disciplines and countries. The authors note that retractions are more frequent in the life sciences and highlight the necessity for careful interpretation of retraction data, as they do not always indicate misconduct, thereby providing a valuable resource for understanding scientific practices and enhancing research evaluation. I think this is an important paper that warrants publication. However, it could be improved in a number of ways, which I outline below. I also wonder if this paper might be better suited for PLOS ONE (although it would not qualify in my mind as suitable with only minor revisions, hence my lack of response to the earlier question of whether it would be "suitable for another PLOS journal with only minor revisions"), given its application to a wide array of scientific fields.

Broadly, my concern is that I didn't get a clear understanding of the purpose of the paper. Is it supposed to elucidate the extent to which highly-cited authors have their papers retracted (and associated variables such as field, region, etc.), describe a newly-published dataset, or both? At present it reads like it is straddling the line between the two, which makes it somewhat difficult to follow. This could, perhaps, be improved by adding a simple statement outlining the purposes of the paper explicitly, early on, but I think the paper would benefit from a more thorough revision that makes the purpose clear at every stage.

I was also somewhat confused by the focus in various places on responsibility on the part of authors. The Method section might benefit from a brief explication of the authors' intentions with the filtering they applied. I presume the goal is to limit the analyses to instances in which the author(s) in question are responsible for errors or malfeasance, but then on p. 10 the authors state that they "make no judgment calls in our databases on the ethical nature of the retractions"; but then, do they not - at least implicitly - do so in the paper? Or am I assuming too much? This is also complicated (as the authors note in multiple instances) by the fact that many of the authors they identify may not be responsible for the elements of the papers that justified their retraction.

A few minor points:

 - The authors note on p. 9 that retracted works often continue to be cited after they have been retracted. This is certainly problematic to the extent that the citing authors are unaware of the retraction, but there are also valid reasons to knowingly cite a retracted work (e.g., to discuss the nature/implications/etc. of the retraction).

 - The authors discuss paper mills in a number of places. A definition would be helpful.

 - The authors mention that some authors may be able to "game the system" re: publishing. An example would be helpful.

 - The authors note that retractions are more common in the life sciences and note that this might be due to increased scrutiny in these fields. This should probably be stated as a higher percentage, as a simple higher rate could be due to base rates (this is reflected elsewhere in the manuscript - just thinking it should be stated as a higher percentage here).

---

## [Editor Report · Decision Letter 2]

12 Dec 2024

Dear John,

Thank you for your patience while we considered your revised manuscript "Retractions among highly-cited authors in science-wide author databases" for publication as a Meta-Research Article at PLOS Biology. This revised version of your manuscript has been evaluated by the PLOS Biology editors and the Academic Editor

Based on our Academic Editor's assessment of your revision, we are likely to accept this manuscript for publication, provided you satisfactorily address the following data and other policy-related requests.

IMPORTANT - please attend to the following:

a) Please change your Title to something more explicit, including an active verb. We suggest the following: "Linking citation and retraction data reveals the demographics of scientific retractions among highly-cited authors"

b) You say that you received no specific funding for this work. Can you please confirm that this is indeed the case?

c) Please address my Data Policy requests below; specifically, we need you to supply the numerical values underlying Figs 2AB, as a supplementary data file.

d) Please cite the location of the data clearly in the legend to Figure 2, e.g. “The data underlying this Figure can be found in S1 Data.”

e) The Academic Editor wants you to include some RRIDs to improve long-term "findability" of the information. Specifically, they suggest the following instances: "To add the new information on retractions, we depended on the most reliable database of retractions available to date, the Retraction Watch database (RWDB, RRID:SCR_000654) which is also publicly freely available through CrossRef (RRID:SCR_003217)." and "Following this filtering process, we linked the retraction records to Scopus (RRID:SCR_022559) using the digital object identifier (DOI) of the original paper..."

f) Where you say "...publications (p<0.001 by Mann-Whitney U..." please report:

• which tool you ran your stats with (and the RRID and version of the tool)

• U-statistic (or z-statistics for large groups), exact p-values, sample and group sizes

• effect size

• descriptive statistics

g) There's a typo in some of the new text ("Carrer-long impact counts" instead of "Career-long impact counts").

h) Please make any custom code available, either as a supplementary file or as part of a DOI'd data deposition (e.g. in Zenodo). For example, I see that you describe the linkage of RetractionWatch entries to Scopus as being automated, so there is presumably a pipeline that performed this linkage? It would also be helpful if a more detailed description of how this linkage was performed were included in the manuscript itself.

We expect to receive your revised manuscript within two weeks. 

*Published Peer Review History*

*Press*

Sincerely,

Roli

Roland Roberts, PhD

Senior Editor

rroberts@plos.org

PLOS Biology

DATA POLICY:

Regardless of the method selected, please ensure that you provide the individual numerical values that underlie the summary data displayed in the following figure panels as they are essential for readers to assess your analysis and to reproduce it: Fig 2AB. NOTE: the numerical data provided should include all replicates AND the way in which the plotted mean and errors were derived (it should not present only the mean/average values).

CODE POLICY

DATA NOT SHOWN?

---

## [Editor Report · Decision Letter 3]

2 Jan 2025

Dear John,

Happy New Year! Thank you for the submission of your revised Meta-Research Article "Linking citation and retraction data reveals the demographics of scientific retractions among highly-cited authors" for publication in PLOS Biology. On behalf of my colleagues and the Academic Editor, Anita Bandrowski, I'm pleased to say that we can in principle accept your manuscript for publication, provided you address any remaining formatting and reporting issues. These will be detailed in an email you should receive within 2-3 business days from our colleagues in the journal operations team; no action is required from you until then. Please note that we will not be able to formally accept your manuscript and schedule it for publication until you have completed any requested changes.

Best wishes,

Roli

Senior Editor

PLOS Biology

rroberts@plos.org